# Combinatorial immunotherapies overcome MYC-driven immune evasion in triple negative breast cancer

Joyce V. Lee[1,2,17], Filomena Housley[1,2,17], Christina Yau[3,4], Rachel Nakagawa[1,2], Juliane Winkler [1,2], Johanna M. Anttila [5], Pauliina M. Munne [5], Mariel Savelius[5], Kathleen E. Houlahan[6], Daniel Van de Mark[1,2], Golzar Hemmati[1,2], Grace A. Hernandez[1,2], Yibing Zhang[1,2], Susan Samson[2,7], Carole Baas[8], Marleen Kok[9,10], Laura J. Esserman [2,11,12], Laura J. van 't Veer [2,4], Hope S. Rugo [2,13], Christina Curtis[6,14,15], Juha Klefström [5], Mehrdad Matloubian[16,18✉] & Andrei Goga [1,2,13,18✉]

Few patients with triple negative breast cancer (TNBC) benefit from immune checkpoint inhibitors with complete and durable remissions being quite rare. Oncogenes can regulate tumor immune infiltration, however whether oncogenes dictate diminished response to immunotherapy and whether these effects are reversible remains poorly understood. Here, we report that TNBCs with elevated MYC expression are resistant to immune checkpoint inhibitor therapy. Using mouse models and patient data, we show that MYC signaling is associated with low tumor cell PD-L1, low overall immune cell infiltration, and low tumor cell MHC-I expression. Restoring interferon signaling in the tumor increases MHC-I expression. By combining a TLR9 agonist and an agonistic antibody against OX40 with anti-PD-L1, mice experience tumor regression and are protected from new TNBC tumor outgrowth. Our findings demonstrate that MYC-dependent immune evasion is reversible and druggable, and when strategically targeted, may improve outcomes for patients treated with immune checkpoint inhibitors.

[1] Department of Cell and Tissue Biology, University of California, San Francisco, California, USA. [2] Helen Diller Family Comprehensive Cancer Center, University of California, San Francisco, California, USA. [3] Cancer and Developmental Therapeutics Program, Buck Institute for Research on Aging, Novato, California, USA. [4] Departments of Pathology and Laboratory Medicine, University of California, San Francisco, California, USA. [5] Finnish Cancer Institute, FICAN South Helsinki University Hospital & Translational Cancer Medicine, Medical Faculty, University of Helsinki, Helsinki, Finland. [6] Stanford Cancer Institute, Stanford University School of Medicine, Stanford, California, USA. [7] Breast Science Advocacy Core, UCSF Breast Oncology Program, San Francisco, California, USA. [8] Alamo Breast Cancer Foundation, San Antonio, Texas, USA. [9] Department of Medical Oncology, Netherlands Cancer Institute, Amsterdam, The Netherlands. [10] Department of Tumor Biology & Immunology, Netherlands Cancer Institute, Amsterdam, The Netherlands. [11] Department of Surgery, University of California, San Francisco, California, USA. [12] Department of Radiology, University of California, San Francisco, California, USA. [13] Department of Medicine, Division of Hematology/Oncology, University of California, San Francisco, California, USA. [14] Department of Medicine, Division of Oncology, Stanford University School of Medicine, Stanford, CA, USA. [15] Department of Genetics, Stanford University School of Medicine, Stanford, California, USA. [16] Department of Medicine, Division of Rheumatology and Rosalind Russell/Ephraim P Engleman Rheumatology Research Center, University of California, San Francisco, California, USA. [17] These authors contributed equally: Joyce V. Lee, Filomena Housley. [18] These authors jointly supervised this work: Mehrdad Matloubian, Andrei Goga. ✉email: mehrdad.matloubian@ucsf.edu; andrei.goga@ucsf.edu

Breast cancer growth is linked to changes in the tumor immune microenvironment and immune escape[1,2]. During immune evasion, the presence of immune checkpoint blockade molecules, such as PD-1 and PD-L1, reduce the tumor killing activity of CD8+ T-cells[2]. PD-L1 is frequently found on tumor cells and/or tumor-associated immune cells in patients with triple negative breast cancer (TNBC)[3,4]. Although patients with PD-L1 positive metastatic TNBC, treated with the checkpoint inhibitor atezolizumab combined with chemotherapy experienced improved overall survival compared to those receiving chemotherapy alone, median survival was still just over 2 years, and there was only a modest improvement in progression free survival (PFS)[5]. This observation suggests that mechanisms other than those mediated by the PD-L1/PD-1 pathway contribute to immune evasion and worse outcomes in TNBC.

Recent studies have focused on assessing immune gene signatures or tumor mutation burden as predictors for immune checkpoint inhibitor response in many cancer types[6–11]. As a complementary approach, we explored whether tumor cell-intrinsic oncogenic drivers could predict patient response to immunotherapy. While our understanding of how oncogenes shape the immune composition of a tumor is growing[12], few studies have directly tested the efficacy of immunotherapies in the context of specific oncogenes[13–15]. We postulated that expression of specific oncogenes may be predictive of response to immunotherapy, which may guide strategies to improve immunotherapy efficacy. To explore this idea, we studied how expression of the oncoprotein c-MYC (MYC) affects immune checkpoint inhibitor response in TNBC.

MYC is frequently overexpressed in TNBC[16–19] and plays a role in tumor recurrence, metastasis, and chemotherapy response[16,20–23]. In lung and pancreatic cancer models, MYC is associated with immune suppression[24,25]. Whether MYC overexpression contributes to immune evasion in TNBC is not clear. In this study, we investigate how MYC shapes the breast tumor immune microenvironment and response to immune checkpoint inhibitors. We hypothesize that poor efficacy of anti-PD-L1 therapy in TNBC is linked to MYC-driven tumor cell immune evasion. We present a clinically viable strategy to reverse MYC-dependent evasion of the immune system, which improves outcomes in MYC-driven TNBC mouse models.

## Results

**MYC-elevated tumors are less sensitive to anti-PD-L1 in preclinical models.** To understand how MYC might alter response to immune checkpoint inhibitors, we initiated our studies using a genetically engineered mouse model of triple negative breast cancer (MTB/TOM)[26], where MYC expression in breast epithelial cells can be switched on and off with doxycycline. Tumors that arose from this model can be propagated via transplantation into the fourth mammary fat pad of wild-type FVBN mice. MYC expression was detected in the tumors while mice were fed doxycycline (MYC-ON) and MYC is no longer detectable in the tumor within 3 days of removing doxycycline from the diet (MYC-OFF) (Supplementary Fig. 1a).

A prior study found that MYC upregulates tumor cell PD-L1, a cell surface molecule that dampens the adaptive immune response, in a mouse model of MYC-driven lymphoma[27], suggesting blocking PD-L1 therapy might be effective in MYC-driven cancers. However, in a combined KRas$^{G12D}$ and MYC-driven lung cancer mouse model anti-PD-L1 treatment was ineffective[24]. These conflicting results prompted us to investigate the role of PD-L1 in MYC-driven TNBC and other cancer types.

Using the MTB-TOM model we allowed each animal's tumor to grow to 10 mm in length and began anti-PD-L1 treatment (Supplementary Fig. 1b). As a single agent, anti-PD-L1 failed to slow tumor growth (Fig. 1a) or to extend survival in mice (Fig. 1b). To test whether increased MYC expression reduced responsiveness to immune checkpoint blockade in another tumor model, we used the anti-PD-L1 sensitive MC38 model, a C57BL6-derived murine colon adenocarcinoma cell line[28] and initiated treatment at 5 mm in length (Fig. 1c, Supplementary Fig. 1c). Compared to control vector transduced MC38 tumors, MYC overexpression was sufficient to decrease anti-PD-L1 efficacy (Fig. 1d). Thus, in two distinct models MYC overexpression renders tumors resistant to anti-PD-L1 therapy.

To test if diminishing MYC expression improves response to anti-PD-L1, we returned to the conditional MYC-driven model of TNBC (MTB/TOM). When tumors grew to 10 mm in length, we removed doxycycline from their diet and concurrently started anti-PD-L1 or isotype control antibody treatment (Supplementary Fig. 1d); the animals remained off doxycycline through the end of the study. Tumors shrank initially, but all eventually recurred spontaneously (Fig. 1e). Nevertheless, by combining MYC inactivation with anti-PD-L1 therapy, we significantly delayed tumor recurrence, extending median survival by ~25% compared to the isotype treated group (69 days in MYC-OFF + isotype antibody vs. 88 days in MYC-OFF + anti-PDL1) (Fig. 1f).

To characterize PD-L1 expression in the tumors, we dissociated MYC-ON tumors and used flow cytometry. While cell surface PD-L1 expression was observed on the tumor-associated myeloid cells (CD45+, monocytes and dendritic cells), it was absent on the tumor cells (CD45−, EPCAM+) (Fig. 1g). MYC inactivation increased PD-L1 expression on tumor infiltrating CD11b+Ly6G− myeloid cells but did not significantly alter PD-L1 expression on CD11b+Ly6G+ neutrophils, on CD11c+ dendritic cells, or on tumor cells (Supplementary Fig. 1e). Taken together, MYC-driven breast tumor epithelial cells express very little PD-L1 and do not respond to anti-PD-L1 monotherapy. While MYC overexpression reduces response to anti-PD-L1, MYC inactivation in combination with anti-PD-L1 delayed tumor recurrence and extended survival in mice, suggesting that MYC expression enhances tumor immune evasion and reduces efficacy of PD-1 blockade as a monotherapy.

**MYC signaling is correlated with poor immune infiltration in patients.** To determine if MYC signaling is correlated with specific immune cell types, which might impact response to anti-PD-L1, we derived a MYC gene signature specific to breast cancer using published gene expression data from multiple MYC-driven mouse models of breast cancer. First, we selected the significantly up- and downregulated genes in the MYC expression subtype (MYC$^{ex}$) derived from the TgMMTV-Myc mouse and TgWAP-Myc mouse from Pfefferle and colleagues[29], and the genes that were significantly altered by MYC in the MTB/TOM model published by our lab[30] (Supplementary Fig. 2a). We identified 530 shared mouse genes regulated by MYC in breast cancer. These mouse genes were then matched to their corresponding human gene IDs to generate an in vivo MYC-driven breast cancer (MYC_BC) signature (Supplementary Data 1). We confirmed that the MYC_BC signature is highly correlated with a previously published in vitro-derived MYC signature[16,31] (Pearson's $R = 0.84$) (Supplementary Fig. 2b) and the Molecular Signatures Database (MSigDB) Hallmark MYC_V1 Targets signature[32] (Pearson's $R = 0.83$) (Supplementary Fig. 2c). However, since our MYC_BC signature was derived from MYC-driven breast tumor models, we anticipated that it would provide additional insight into MYC-driven tumor-immune alterations relevant to TNBC.

We used the MYC_BC signature to explore TNBC tumor gene expression from the Cancer Genome Atlas (TCGA) dataset.

Notably, in the TCGA TNBC patient cohort, those with a high MYC_BC signature had less *CD274* expression, the gene that encodes for PD-L1 (Pearson's $R = -0.43$) (Fig. 1h). Next, we explored how the MYC_BC signature correlated with published immune cell signaling signatures[33]. Overall, tumors with a high MYC_BC signature were associated with reduced tumor infiltrating leukocytes (TILs) signature[33] (Pearson's $R = -0.63$) (Fig. 1i). Additionally, MYC activation was negatively correlated with multiple other published immune cell type signatures[33], associated with fewer T cells, B cells, macrophages, and NK cells (Supplementary Fig. 3, Supplementary Fig. 4). The MYC signature also correlated with low T-cell cytokine signaling (HALLMARK IL2/STAT5)[32] in the TCGA TNBC patient cohort (Pearson's $R = -0.74$) (Fig. 1j). We further corroborated our findings of an inverse relationship between MYC and low immune signatures in the METABRIC[34,35] TNBC patient populations (Fig. 1k, Supplementary Fig. 4) and in the TONIC trial, an immunotherapy trial for patients with metastatic TNBC[36] (Fig. 1l).

**MYC associates with poor survival after immune checkpoint inhibition.** Given the low immune signatures associated with MYC-elevated human tumors we predicted that for patients treated with immune checkpoint inhibitors, those whose tumors have a high MYC signature score would have worse outcomes. In the TONIC Trial patients were randomized to receive nivolumab (anti-PD-1) in combination with different induction chemotherapies (cyclophosphamide, doxorubicin, irradiation, or cisplatin)[36]; the MYC_BC signature was associated with the non-responders regardless of induction therapy (Supplementary Fig. 5). Additionally, we looked at a small cohort of patients with early stage TNBC treated with pembrolizumab (anti-PD-1) plus standard chemotherapy in the neoadjuvant setting from the I-SPY 2 Trial[37]. At the median follow up time of 2.4 years, all observed recurrences were among the patients with a high MYC_BC signature, and no recurrence was seen in those patients with a low MYC_BC signature (Fig. 1m). Though both studies are underpowered, the clear trend pointed to MYC as a potential predictive biomarker to evaluate outcomes of patients receiving immune checkpoint inhibitors. We were further prompted to look at other published immunotherapy datasets through the Tumor Immune Dysfunction and Exclusion (TIDE) platform[38,39]. Because the database does not have trials involving breast cancer, we queried for *MYC* expression rather than using the MYC_BC signature. Urothelial cancer[40] and renal cell carcinoma (ccRCC)[41] are tumor types with frequent MYC amplification[42,43] or overexpression[42,44]. In urothelial cancer, patients had shorter median overall survival after anti-PD-L1 treatment if their pre-treatment tumors expressed high *MYC* ($p = 0.0112$) (Fig. 1n). A trend toward diminished overall survival was also found with elevated *MYC* in patients with clear cell renal cancers (ccRCC; $p = 0.0578$) (Fig. 1o). Thus, both in our preclinical models and analysis of patient datasets, we find that MYC correlates with low T-cell activation and poor responses to immune checkpoint blockade.

**MHC-I is downregulated in MYC-activated tumors.** Though PD-L1 was present on tumor-associated immune cells in the MYC-driven TNBC model (Fig. 1g), the poor tumor response to anti-PD-L1 (Fig. 1a, b) led us to postulate that other factors might explain the observed immune evasion. This prompted us to explore the cellular programs that could contribute to poor response in the MYC-ON state. Tumor-specific antigen presentation by MHC class 1 (MHC-I) is necessary for T-cell recognition and cytotoxicity[45]. Somatic loss of heterozygosity in antigen presentation machinery is associated with resistance to immunotherapy in patients with breast cancer[23], melanoma[46] and lung cancer[47,48]. Likewise, genetic deletion of a key MHC-I component, β2-microglobulin (*B2m*), abrogates the therapeutic effect of anti-PD-1 in mouse melanoma[49]. We decided to explore the expression of *B2M* and other antigen presentation genes in the context of MYC overexpression and immunotherapy. Although a link between MYC family proteins and MHC-I heavy chain gene expression was described over 30 years ago in neuroblastoma and melanoma cell lines[50,51], this link has not been explored in breast cancer, and its significance for immune checkpoint inhibitor response has not been investigated in human patient samples or in vivo cancer models.

TNBC patients with a high MYC_BC signature had lower expression of genes important for MHC-I expression (Supplementary Fig. 6), such as *B2M* (Fig. 2a, b) and *NLRC5* (Fig. 2c, d), the major transactivator of MHC-I genes[52,53]. We further validated that MYC overexpression is accompanied by a loss of MHC-I gene expression in human cells by examining gene expression in the immortalized, non-tumorigenic human mammary epithelial cells. MCF10A cells overexpressing ectopic MYC[54] displayed lower expression of antigen presentation machinery genes than parental MCF10A cells, indicating that MYC overexpression is sufficient to drive the repression of MHC-I related genes in human breast epithelial cells (Fig. 2e).

Next, we compared the expression of antigen presentation genes in the MYC-ON mouse breast tumors to normal mouse mammary glands (Ctrl). The presence of MYC significantly downregulated multiple genes important for antigen presentation by MHC-I (Fig. 2f) compared to normal mammary epithelium. Remarkably, most antigen presentation genes were re-expressed within 3 days of inactivating MYC (MYC-OFF) (Fig. 2f), demonstrating that the expression of antigen presentation genes are reversible with downregulation of MYC. We investigated whether MYC exhibited similar effects on MHC-I genes in conditional transgenic models of liver cancer[55] and lymphoma;[30] indeed, turning MYC off in these models also resulted in an increase in antigen presentation genes (Supplementary Fig. 7a, b), suggesting that the downregulation of antigen presentation genes is dependent on MYC in multiple conditional transgenic cancer models.

To validate the reduction of MHC-I cell surface expression, we used flow cytometry on intact cells to evaluate MHC-I in MTB/TOM tumors after MYC inactivation. Tumor cells (CD45−, EPCAM+) in the MYC-OFF state displayed significantly more surface MHC-I protein than MYC-ON tumor cells (Fig. 2g). We also observed increased tumor infiltration of CD8+ T cells in the MYC-OFF state, while the infiltration by other immune subtypes was not significantly altered (Fig. 2h, i). Together, these data demonstrate MYC-ON tumors have low MHC-I expression and turning MYC off rescued MHC-I cell surface expression and concurrently increased CD8+ T-cell tumor infiltration. These two phenotypes observed upon MYC inactivation provide a possible explanation as to why low MYC conditions resulted in better responses to immune checkpoint blockade in Fig. 1a–f. Furthermore, these studies demonstrate that the mRNA and cell surface expression of MHC-I is reversible and thus, targetable.

**Interferons rescue MHC-I in MYC-elevated tumors.** Without any approved effective drugs that directly inhibit MYC, we dug deeper into the MTB/TOM RNA-Seq data for clues toward targeting MYC-driven breast cancers in vivo. We identified 1328 genes that were downregulated in the MYC-ON state but upregulated when MYC is acutely turned off (Supplementary Fig. 8). Using GSEA/MSigDB[32,56] to characterize these genes, we found

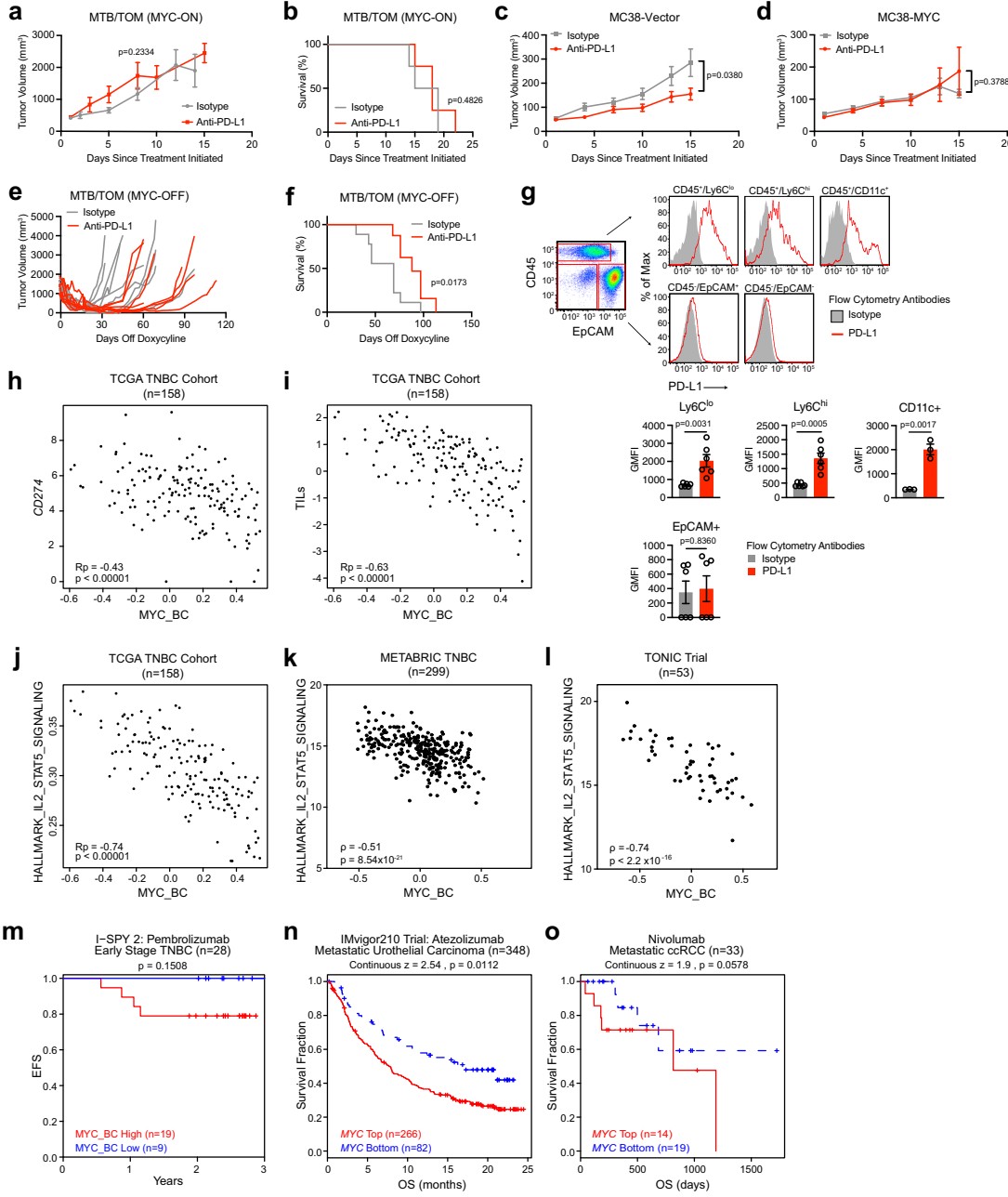

**Fig. 1 MYC predicts poor response to immune checkpoint inhibitors. a** Average tumor volume for MTB/TOM tumors treated with anti-PD-L1 or isotype antibody while fed doxycycline chow (MYC-ON state). Mean ± S.E.M; two-sided unpaired $t$-test on day 8, isotype ($n = 3$), anti-PD-L1 ($n = 4$). **b** Survival (ethical tumor endpoint, length: 20 mm). Log rank test, isotype ($n = 4$), anti-PD-L1 ($n = 8$). **c** Average tumor volumes for animals bearing MC38-vector tumors. Mean ± S.E.M, two-sided unpaired $t$-test tumor volume on day 15 (isotype $n = 8$, anti-PD-L1 $n = 10$). **d** Average tumor volumes for animals bearing MC38-MYC tumors. Mean ± S.E.M, two-sided unpaired $t$-test tumor volume on day 15 ($n = 6$/treatment). **e** Individual tumor volumes graphed during doxycycline chow removal (MYC-OFF) and either anti-PD-L1 ($n = 8$) or isotype control antibody ($n = 9$). **f** Survival (ethical study endpoint). MYC-OFF + isotype antibody ($n = 9$) and MYC-OFF + anti-PDL1 ($n = 8$). Log rank test. **g** Top: Representative flow cytometry plots. PD-L1 expression (red) or isotype antibody (gray) displayed as percent of maximum (modal). Bottom: Bar graphs displaying average geometric mean fluorescence intensity (GMFI) for PD-L1 in each population. $n = 6$ tumors analyzed for each cell type, except for CD11c ($n = 3$), mean ± S.E.M., unpaired $t$-test. **h–j** Scatterplots of **h** *CD274*, **i** tumor infiltrating leukocytes (TIL), and **j** IL2/STAT5 Hallmark gene signature against the MYC_BC signature in the TCGA TNBC cohort ($n = 158$). Pearson's coefficient (Rp), adjusted $p$-value (Benjamini–Hochberg FDR corrected). **k** Evaluation of IL2/STAT5 Hallmark gene signature against the MYC_BC signature in METABRIC. ρ, Spearman's $r$ and exact $p$-value displayed. **l** Evaluation of IL2/STAT5 Hallmark gene signature against the MYC_BC signature in the TONIC Trial. ρ, Spearman's $r$ and exact $p$-value displayed. **m** Kaplan–Meier curves of event-free survival for patients with HER2 negative and hormone receptor (HR) negative tumors treated on the Pembrolizumab arm of the I-SPY 2 TRIAL dichotomized by MYC_BC signature. Cox proportional hazard (PH) modeling. **n, o** Survival analysis correlating *MYC* expression from pre-treatment tumors pulled out of the TIDE database using Cox PH z-scores: **n** Kaplan–Meier curves of overall survival in patients with metastatic urothelial carcinoma from Mariathasan et al., 2018 and **o** Kaplan–Meier curves of overall survival in patients with metastatic clear cell renal cell carcinoma (ccRCC) from Miao et al., 2018. Source data are provided in the Source Data file. Source data for panel l provided at EGA (EGAS00001003535).

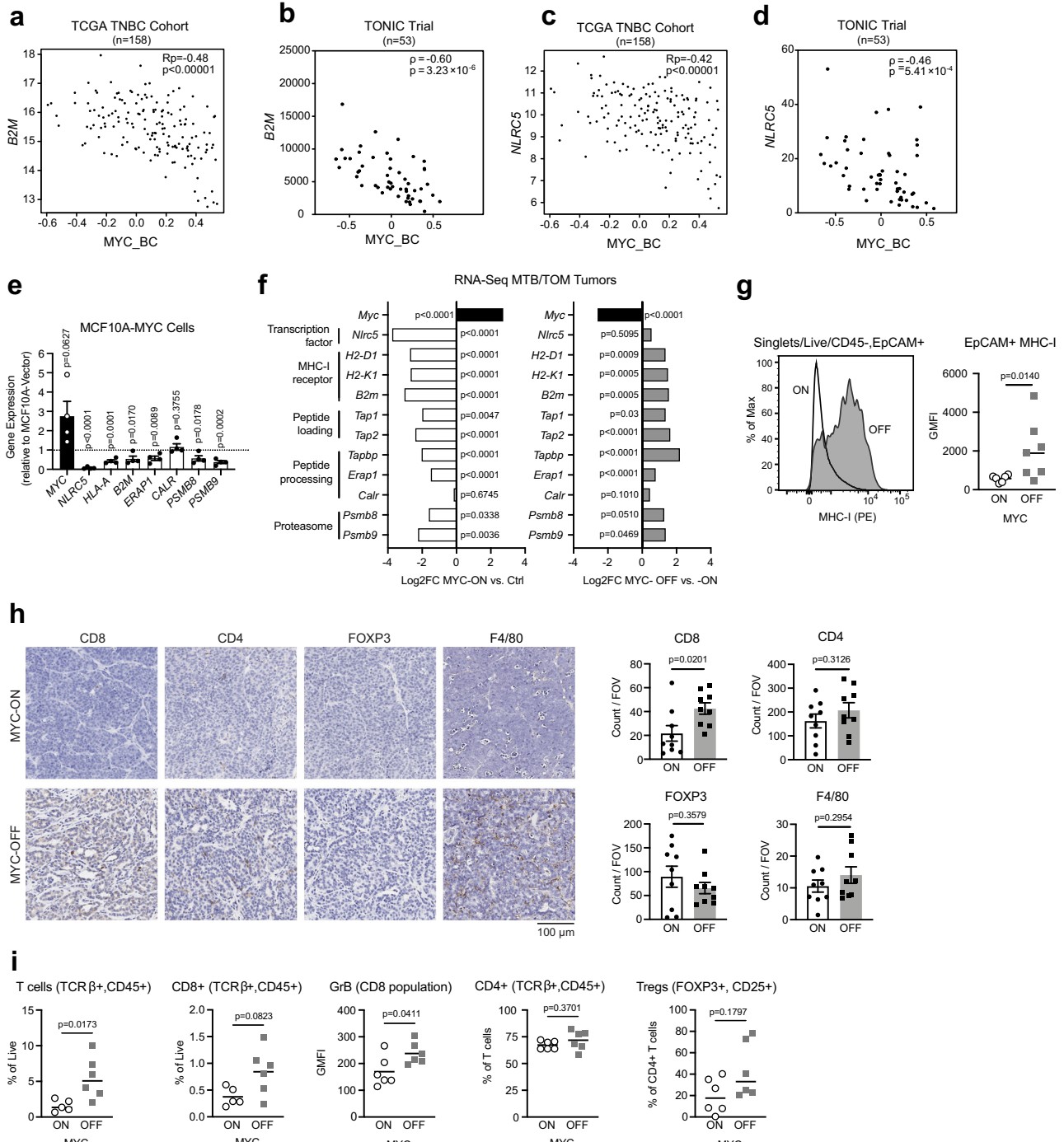

**Fig. 2 MYC suppression of MHC-I expression is reversible. a–d** Scatter plot correlation tested between the expression of (**a**) *B2M* gene and MYC_BC signature in TCGA TNBC patients, Pearson's coefficient (Rp), adjusted *p*-value (Benjamini-Hochberg FDR corrected). **b** *B2M* and the MYC_BC signature in TONIC Trial patients, Spearman's r (ρ), exact *p*-value. **c** *NLRC5* gene and MYC_BC signature in TCGA TNBC patients, Pearson's coefficient (Rp), adjusted *p*-value (Benjamini–Hochberg FDR for corrected). **d** *NLRC5* gene and the MYC signature in the TONIC Trial patients, Spearman's r (ρ), exact *p*-value.
**e** Antigen presentation pathway genes in MCF10A-MYC cells compared to MCF10A-vector cells. Data displayed as fold change with values < 1 (dotted line) equating to gene repression. Displayed *n* = 4 independent cell passage replicates. Mean ± S.E.M, two-sided, unpaired *t*-test, exact *p*-values.
**f** Differential expression analysis for RNA-seq data in MTB/TOM tumors published by Rohrberg et al. Left: MYC-ON compared to normal mammary tissue (Ctrl), Right: 3 days off-doxycycline (MYC-OFF) compared to animals on doxycycline (MYC-ON). Adjusted *p*-values. **g** Representative histogram for MHC-I expression by flow cytometry in MYC-ON and MYC-OFF states. Adjacent bar graph displaying geometric mean fluorescence intensity (GMFI) for MHC-I in MYC-ON (*n* = 6) and MYC-OFF (*n* = 7). Line represents median, data points represent individual animals, two-sided, Mann–Whitney test, exact *p*-value.
**h** Immunohistochemistry staining of immune cell markers within tumor sections in MYC-ON and MYC-OFF state. Images are at 20X. Adjacent bar graphs displaying cumulative counts per field of view (FOV). *n* = 3 animals per group, 3 FOV analyzed per tumor, mean ± S.E.M. two-sided unpaired *t*-test, exact *p*-values. **i** Flow cytometry analysis of immune cells found in MYC-ON tumors or MYC-OFF tumors (off doxycycline chow for 3 days) (*n* = 6 per group). Line represents median, data points are individual animals, outliers removed, two-sided Mann–Whitney test, exact *p*-values. Source data are provided in the Source Data file. Source data for panels b and d provided at EGA (EGAS00001003535).

that the top MYC-downregulated pathways were enriched for immune processes (Supplementary Table 1). These findings raised the possibility that restoring immune infiltration could improve efficacy of anti-PD-L1 therapy. The interferon response pathways were among the most suppressed pathways in MYC-activated tumors (Supplementary Table 1). We examined the hallmark interferon-alpha and interferon-gamma pathways in TNBC tumors; high MYC signature was strongly associated with less interferon signaling in TCGA (Fig. 3a), in METABRIC (Fig. 3b), and in the TONIC trial (Fig. 3c).

Canonically, MHC-I is expressed robustly upon interferon signaling[57], but whether this signaling still occurs in MYC-elevated breast cancers was unknown. We tested whether MTB/TOM cells could be induced to express MHC-I after exogenous interferon stimulation. We exposed a cell line derived from MTB/TOM tumors to type I interferons (IFNα or IFNβ) or type II interferon (IFNγ), while in the MYC-ON state. Following 72 h exposure, we observed the induction of several antigen presentation genes (Fig. 3d), with IFNγ inducing a greater change than the type I interferons (Fig. 3e), and a dramatic increase of MHC-I protein on the cell surfaces of the MTB/TOM tumor cells (Fig. 3f). We tested whether the re-expression of MHC-I was due to changes in MYC abundance. We did not observe changes in MYC protein expression by IFNα treatment (Fig. 3g), but noted a slight downregulation in MYC after IFNγ treatment (Fig. 3h). These studies demonstrate that interferon treatment increases MHC-I surface expression on MYC-elevated tumor cells, but this is not solely dependent on MYC downregulation.

**CpG/aOX40 treatment improves anti-PD-L1 efficacy.** Activation of pattern recognition receptor pathways, such as toll-like receptor 9 (TLR9), leads to the local production of interferons in the tumor microenvironment. A synthetic TLR9 agonist, unmethylated CpG oligodeoxynucleotides (CpG-ODN, hereafter CpG), can stimulate plasmacytoid dendritic cells (pDC) to produce IFNα and IFNβ in the local microenvironment, activating both B and T cells and attracting natural killer (NK) cells; this cascade upregulates production of IFNγ and subsequently can attract antigen-specific CD8+ T-cells[58]. We asked whether a concerted interferon response induced by CpG would increase MHC-I expression in the MYC-driven mouse mammary tumors. After a single intratumoral administration of CpG directly into MYC-ON tumors, we observed more MHC-I expression on the tumor cells (Fig. 4a, Supplementary Fig. 9). CpG did not decrease MYC protein abundance (Supplementary Fig. 10), which suggests that the mechanism to induce MHC-I does not depend on lowering MYC expression in vivo.

Intratumoral CpG extended survival, but CpG combined with systemic anti-PD-L1 did not extend survival beyond CpG alone (Supplementary Fig. 11), suggesting another form of immune activation is required. A recent study revealed that CpG induces the expression of OX40, a co-stimulatory molecule, on CD4+ T-cells (including suppressive T-regulatory cells, Tregs)[59]. Stimulation of Tregs through OX40 impairs their function, which is critical for tumor shrinkage in the spontaneous polyomavirus middle T-antigen breast cancer mouse model[59]. We decided to test whether a combination of CpG and an agonistic antibody against OX40 (aOX40), could delay tumor progression in MTB/TOM tumors. We administered intratumoral injections of CpG oligo and aOX40 (CpG/aOX40) in MYC-ON tumors (5 mm) every other day for a total of 3 injections and then monitored tumors to study endpoint. CpG plus aOX40 treatment delayed tumor progression and increased median survival (56.5 days for CpG/aOX40 group vs. 30 days for the isotype/vehicle control group) (Fig. 4b).

To discern the effects of CpG/aOX40 on the tumor microenvironment, we examined the immune composition at day 10, when the anti-tumor effects of CpG/aOX40 emerged (Fig. 4c). We detected a dramatic increase in immune infiltration as detected by immunohistochemistry staining, including total T-cells (CD3+), specifically, CD4+ and CD8+ T-cells, and macrophages (F4/80) (Supplementary Fig. 12). These results were confirmed and further characterized by flow cytometry. We found an increase in T-cell (CD45+, TCRβ+) infiltration in the tumors given CpG/aOX40 compared to tumors given isotype control or aOX40 alone (Fig. 4d). Concurrently, we observed a lower proportion of Tregs, (CD25+, FoxP3+), within the CD4+ T-cell population, for CpG/aOX40 treated tumors (Fig. 4e). Compared to single agent treated tumors, CpG/aOX40 treated tumors displayed a higher CD8+ T-cell to Treg ratio, which is important for the anti-tumor phenotype[60] (Fig. 4f). Furthermore, the CD8+ T-cells (CD45+, TCRβ+/CD8+, CD4−) expressed greater amounts of granzyme B molecules in the CpG group and CpG/aOX40 group, compared to the isotype treated tumors (Fig. 4g), suggesting that the specific treatments increased CD8 T-cell functionality and ability to initiate tumor killing. Overall, CpG/aOX40 treatment sustained high CD8+ T-cell:Treg ratio and granzyme B production, suggesting that this treatment can increase CD8+ T-cell functionality and the ability to initiate tumor killing.

Given both the quantitative and qualitative changes in the intratumoral CD8+ T cells following CpG/aOX40 treatment in MYC-ON tumors, we reasoned that this approach would improve the sensitivity to anti-PD-L1 therapy. We therefore tested CpG/aOX40 together with anti-PD-L1. Once tumors reached 5 mm, animals were randomized into the four treatment arms (Supplementary Fig. 13). Isotype antibody treated and anti-PD-L1 alone treated animals displayed the expected rapid tumor progression (Fig. 4h, i). In the CpG/aOX40 group, 33% of the animals had tumors regressed (Fig. 4h–j). Remarkably, addition of anti-PDL1 (triple combination) resulted in complete and long-term regression of tumors in 75% of the animals, at 100 days post-treatment initiation (Fig. 4h–j).

We tested the triple combination in two additional MYC overexpressing models. MC38-MYC overexpressing tumors, which were resistant to anti-PD-L1 monotherapy (Fig. 1d), now responded to CpG/aOX40 and the triple combination (Fig. 4k). We also tested a different transgenic model of MYC-driven luminal B breast cancer (WAP-MYC)[61] and also found that triple combination therapy (CpG/aOX40 + anti-PD-L1) resulted in improved responses and significantly improved survival (Fig. 4l).

We next asked if tumor-bearing mice with complete regression of their tumors would be resistant to subsequent rechallenge. Mice with fully regressed tumors after triple combination therapy were rechallenged with new MTB/TOM tumor transplants in their contralateral mammary fat pads (left side, fourth gland). FVBN control mice that had not received prior tumor transplantation nor previous therapy served as controls for tumor growth. All animals that responded to the initial therapy successfully eliminated growth of new tumor transplants (Fig. 4m), demonstrating that the combination therapy led to a durable immune response that protected mice from establishing new tumor outgrowth.

## Discussion

Immune checkpoint inhibitors represent promising treatments for patients. Efficacy is seen in patients with PD-L1 positive tumors, but few patients achieve full or durable remission[62]. We observed that anti-PD-L1 was ineffective in MYC-driven breast cancer models, even though PD-L1 was expressed on tumor-

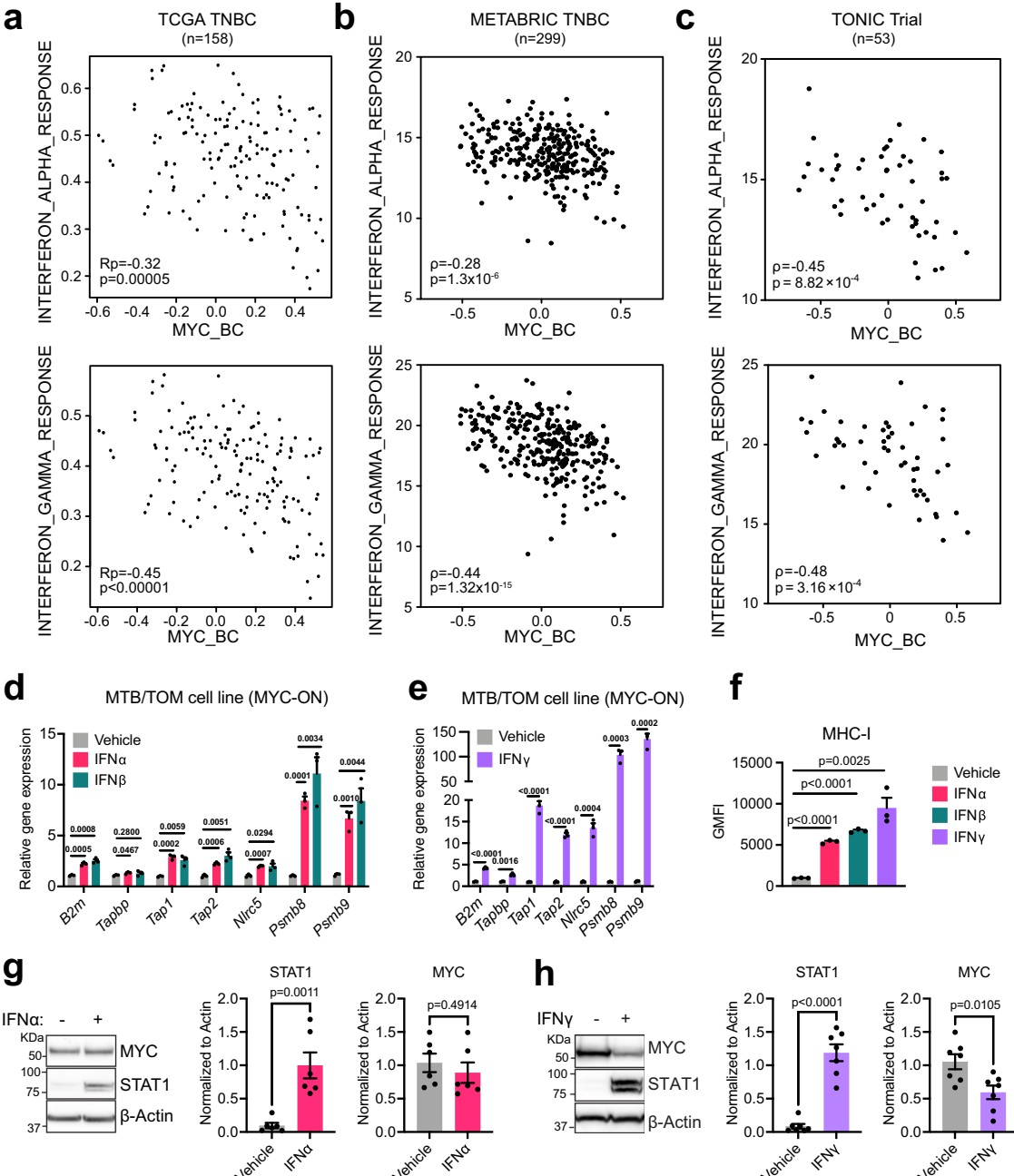

**Fig. 3 Interferon signaling rescues MHC-I in MYC-elevated cells. a–c** Scatter plot correlation tested between the expression of Hallmark Interferon Alpha Response signature or Hallmark Interferon Gamma Response signature and the MYC_BC signature in **a** TCGA TNBC patients, Pearson's coefficient (Rp), adjusted $p$-value (Benjamini–Hochberg FDR corrected); **b** METABRIC TNBC, ρ, Spearman's $r$ and exact $p$-value; and **c** TONIC Trial, ρ, Spearman's $r$ and exact $p$-value. **d**, **e** Gene expression analysis after 72 h of **d** vehicle, interferon alpha, and interferon beta, or **e** interferon gamma treatment in the presence of doxycycline, in MTB/TOM cells grown in vitro culture. Representative experiment with three samples per condition shown. Mean ± S.E.M, unpaired $t$-test with Benjamini–Hochberg FDR = 0.05, adjusted $p$-values displayed across the bars. Trends repeated with three independent cell passages. **f** Flow cytometry results displayed as geometric mean fluorescence intensity (GMFI) for MHC-I in MTB/TOM cell line in vitro culture after 72 h of treatment. Representative experiment with three samples per condition shown. Mean ± S.E.M, two-sided, unpaired $t$-test, exact $p$-values. Trends repeated with three independent cell passages. **g**, **h** Left: Representative western blots showing MYC levels in the MTB/TOM cell line after 72 h of treatment with **g** interferon alpha ($n = 6$ independent cell passages) or **h** interferon gamma ($n = 7$ independent cell passages). Right: densitometry ratio of MYC or STAT1 to ß-actin shown from independent cell passages. Two-sided, unpaired $t$-test, exact $p$-values. Source data are provided in the Source Data file. Source data for panel c provided at EGA (EGAS00001003535).

associated myeloid cells. This highlighted that in the context of MYC, additional tumor immune evasion mechanisms were relevant.

The low efficacy of anti-PD-L1 in MYC-elevated cancers is in part linked to suppression of MHC-I genes and low tumor immune cell infiltration in patient tumors and mouse models.

The MYC-dependent suppression of T-cell infiltration seen here is supported by previous findings in a composite KRas[G12D]/MYC-driven lung cancer model and pancreatic cancer models[24,25]. A recent paper highlighted that TNBC patients with a high copy number value of chromosome 8q24 (which contains the *MYC* locus) tended to possess tumors with a non-inflamed gene

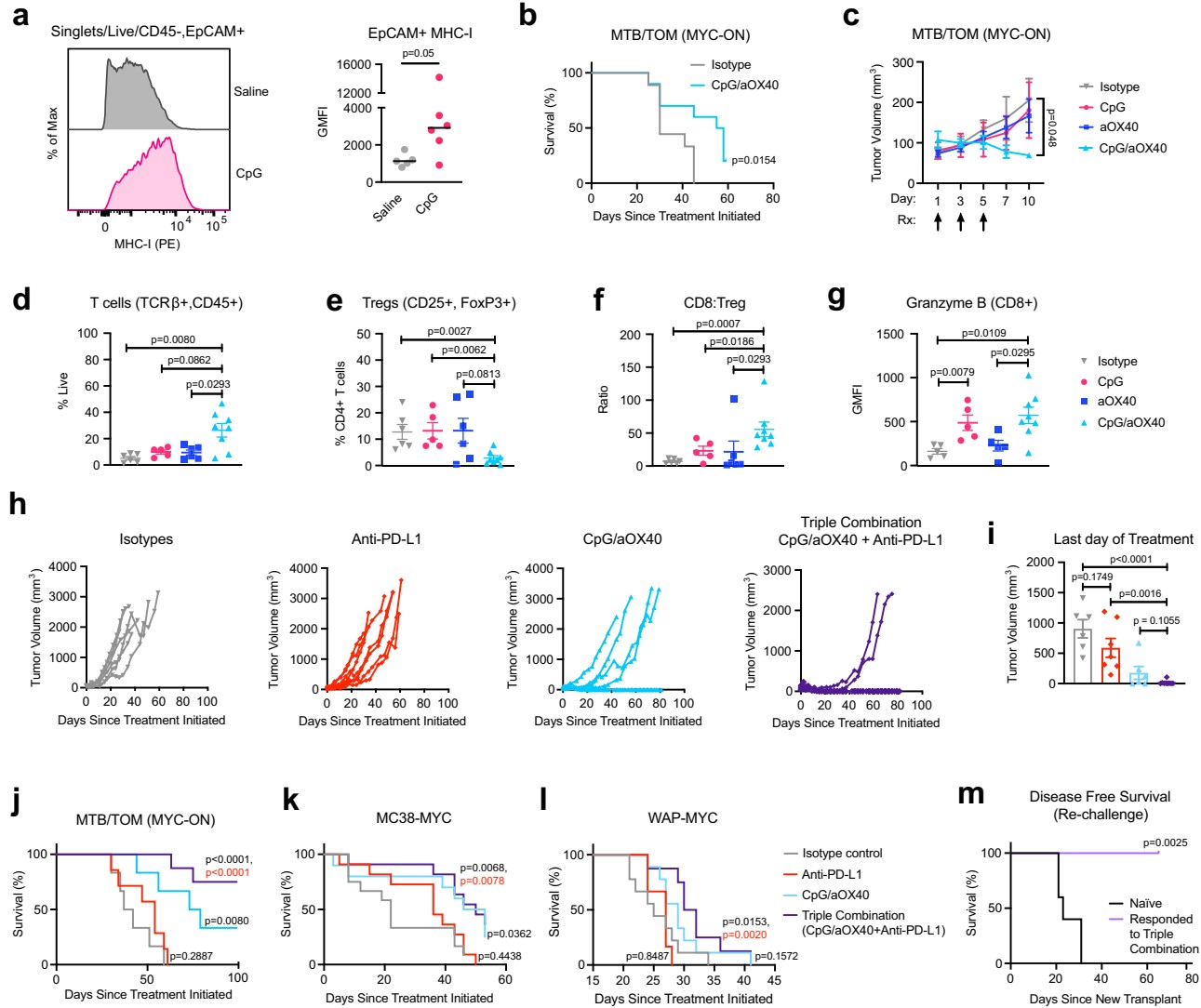

**Fig. 4 CpG/aOX40 enhances anti-PD-L1 in vivo. a** Representative histogram of cell surface MHC-I detected by flow cytometry. Adjacent bar graph displaying geometric mean fluorescence intensity (GMFI). Line represents median, data points represent individual animals, saline ($n = 5$), CpG ($n = 6$), Mann–Whitney test, exact $p$-value. **b** Survival (time to endpoint length: 20 mm) for animals given CpG/anti-OX40 (aOX40) ($n = 10$) or isotype/vehicle ($n = 9$). Log rank test, exact $p$-value. **c** Tumor volume following treatments (Rx) indicated by arrows. Isotype ($n = 6$), CpG ($n = 5$), aOX40 ($n = 6$), CpG/aOX40 ($n = 5$). Mean ± S.E.M, two-sided, unpaired $t$-test comparing tumor volume on day 10 in isotype treated vs. CpG/aOX40, exact $p$-value. **d–g** Flow cytometry analysis of immune cells found in the tumor post-treatment initiation day 10. Isotype ($n = 6$), CpG ($n = 5$), anti-OX40 ($n = 6$), CpG/aOX40 ($n = 8$). Mean ± S.E.M., Mann–Whitney test., exact $p$-values: **d** Percent of live cells in the tumor that are T-cells. **e** Percent of CD4+ (gating: Singlets/Live/TCRβ+, CD45+/CD8−, CD4+) T-cells that are CD25+ and FOXP3+. **f** Ratio of CD8+ (gating: Singlets/Live/TCRβ+, CD45+/CD8+, CD4−) T-cell counts to Treg counts (gating: Singlets/Live/TCRβ+, CD45+/CD8−, CD4+/CD25+, FOXP3+). **g** Geometric mean fluorescence intensity for intracellular granzyme B in the CD8+ T-cells (gating: Singlets/Live/TCRβ+, CD45+/CD8+, CD4−). **h** MTB/TOM tumor volumes graphed to endpoint: isotype ($n = 6$), anti-PD-L1 ($n = 7$), CpG/aOX40 ($n = 6$), and triple combination ($n = 9$). **i** Tumor volume in MTB/TOM animals after last day of treatment (related to panel **h**). Mean ± S.E.M., two-sided, unpaired $t$-test, exact $p$-value. **j–l** Survival (time to ethical endpoint length = 20 mm). Log rank test for each treatment arm in comparison to isotype arm, exact $p$-value in black. Log rank test for anti-PD-L1 compared to triple combination, exact $p$-value in red: **j** MTB/TOM (MYC-ON) animals from panel **h**. **k** MC38-MYC model, isotype ($n = 12$), anti-PD-L1 ($n = 11$), CpG/aOX40 ($n = 10$), triple combination ($n = 11$). **l** WAP-MYC model, isotype ($n = 9$), anti-PD-L1 ($n = 6$), CpG/anti-OX40 ($n = 9$), and triple combination ($n = 8$). **m** Disease-free survival in MTB/TOM model (defined as time to palpable tumor). Mice previously treated with triple combination and eradicated their tumors were transplanted with a new MTB/TOM tumor on the contralateral side. Tumor naïve group ($n = 5$) and responded to therapy group ($n = 5$). Log rank test, exact $p$-value. Source data are provided in the Source Data file.

signature[63], consistent with our findings. *MYC* overexpression or amplification is associated with a basal-like (BL1) TNBC subtype which is associated with proliferation[64]. By examining downregulated rather than upregulated signaling pathways, we discovered that major processes inhibited by MYC overexpression and then restored once MYC is inactivated are related to immune responses. We further demonstrate that inactivation of MYC in

the MTB/TOM TNBC mouse model restored MHC-I cell surface expression, CD8+ T-cell infiltration, and response to anti-PD-L1. Collectively, our data emphasizes that MYC-dependent deregulation of MHC-I and immune exclusion is reversible and thus, druggable by increasing interferon signaling. In lung cancer cells, EZH2 inhibitors[65] and HDAC inhibitors[66] have been demonstrated to restore MHC-I in vitro, suggesting these drugs might

also increase MHC-I in lung tumors. Here, we demonstrate in vivo that administration of a single, low dose of CpG alone was sufficient to bypass MYC-dependent repression of breast tumor cell MHC-I expression, and it increased the fraction of cytotoxic CD8 + T-cells. Importantly, a short interval treatment of CpG/aOX40 significantly reduced the fraction of immunosuppressive Tregs within mouse MYC-driven TNBC tumors and increased both numbers and functionality of tumor infiltrating cytotoxic CD8 + T cells. When combined with anti-PD-L1, a majority of tumors fully regressed and the mice were resistant to further tumor implants at a second site, suggesting that local immunostimulation combined with a systemic checkpoint inhibitor is effective in MYC-driven tumors and can provide durable systemic anti-tumor immunity. CpG/aOX40 was previously demonstrated to be effective in multiple preclinical cancer models[59,67]. Human OX40 agonists and TLR-agonists are currently being evaluated in phase I clinical trials (NCT03831295, NCT03410901). Our data demonstrates MYC-elevated tumors are among those tumors that would benefit from this combination, although the addition of anti-PD-L1 is likely required for optimal efficacy. Future work to explore the spatial distribution of TILs in MYC-high TNBCs and following combinatorial immunotherapies should provide additional insight into mechanisms of immune evasion and how they can be overcome.

Our study suggests that patients with MYC-elevated tumors will likely require additional immune therapies beyond anti-PD1/PD-L1 monotherapy to achieve optimal survival benefit with immune checkpoint inhibitors. This is highlighted in our analysis of event-free survival (EFS) for patients with TNBC in the pembrolizumab arm of the ISPY-2 TRIAL[37], which revealed that patients with a high MYC signature in their pre-treatment tumors experienced tumor relapse or metastasis sooner than patients with a low MYC signature. All patients received chemotherapy together with immune checkpoint blockade, suggesting that chemotherapy is not sufficient to cause an immunogenic response in tumors with high MYC-signaling. MYC may also be an important predictor of outcome for immune checkpoint inhibitor therapy in other tumor types (such as urothelial bladder cancer and ccRCC). Clinical trials like the Phase II I-SPY 2 and the Phase III ILLUMINATE-301 (NCT03445533) testing immune checkpoint inhibitors and TLR9 agonists are ongoing and it will be exciting to see how patients with MYC-elevated tumors respond to these combinations. Our data demonstrate that while MYC challenges immune checkpoint blockade therapies in patients with MYC-elevated tumors, additional therapeutics in combination with anti-PD-L1 are warranted to be tested and would be predicted to improve survival. In future studies, it will be important to also understand the signaling pathways and biomarkers in the small subset of MYC-high tumors that did not respond to triple-combination immune therapies. The implementation of our defined MYC_BC signature based on primary MYC-driven breast tumors may provide a strategy for identifying patients that are at high risk for progression following treatment with immune checkpoint inhibitors. Moreover, we propose that therapeutics that reactivate MHC-I expression and enhance anti-tumor immune cell infiltration in MYC-high tumors may improve the efficacy of immune checkpoint inhibitor therapies like anti-PD-1 and anti-PD-L1.

## Methods
Our research complies with the University of California, San Francisco's regulations for chemical usage (Environmental Health and Safety, Protocol CS107458-04), for biological agent usage (Environmental Health and Safety, Protocol BU038296-05D), and for animal studies (Institutional Animal Care and Use Committee, Protocol AN184330-01; and National Animal Experiment Board of Finland, License ESAVI-2010-05551_Ym-23, KEK19-002).

**Tumor initiation in mice**
*MTB/TOM and MC38*. Viably frozen MTB/TOM tumors (lab lines: A and B), generated from MMTV-rtTA/TetO-MYC mice on the FVBN background, were divided into fragments (~8 mm³) and one piece was implanted into the cleared right 4th mammary fat pad of each 4-week-old female FVBN mice (Jackson Laboratory Stock 001800) under 2.5% isoflurane. Mice were maintained on doxycycline diet (Bio-Serv #S3888) starting 1 day before transplant surgery. For the MC38 tumor studies, 6–7-week-old female C57BL/6 J mice (Jackson Laboratory Stock 000664) under 2% isoflurane were injected with $2 \times 10^5$ cells suspended in 100 μl DMEM subcutaneously in the right flank. All mice imported from Jackson Labs were allowed 3–7 days to acclimate to our facilities prior to transplantation or tumor cell injections. All mice were maintained at UCSF rodent barrier facilities. All procedures were approved by UCSF Institutional Animal Care and Use Committee (IACUC) under protocol number AN184330-01.

*WAP-MYC tumor initiation in mice*. Fat pad clearance and tumor cell transplantations were performed to 4-week-old female FVB-recipient mice. The mice were treated with analgesic 30 min prior the transplantation. The mice were anesthetized using inhaled 2.5% isoflurane. Freshly isolated WAP-MYC tumor cells were used in the transplantations. $1 \times 10^5$ primary WAP-MYC tumor cells in PBS were injected in 10 μl volume into both remaining fat pads. All animals were covered by a license (ESAVI-2010-05551_Ym-23, KEK19-002) approved by the National Animal Experiment Board of Finland (Eläinkoelautakunta, ELLA).

**Tumor studies**. For PD-L1 monotherapy studies, animals were randomized to treatment groups when tumors reached 10 mm; for animals off doxycycline (MYC-OFF), animals were moved into new cages with provided standard chow. For combinatorial therapy experiments, animals were randomized to treatment arms once tumors reached 5 mm in length. Saline was used to dilute the drugs. For each arm: CpG (IDT) was delivered intratumorally with an insulin syringe at 50 μg in 0.05 mL per injection; anti-OX40 (Bio X Cell #BE0031) was delivered intratumorally with an insulin syringe at 8 μg in 0.05 mL; CpG/anti-OX40 together was delivered in a single injection containing 50 μg of CpG and 8 μg of anti-OX40 in 0.05 mL. Anti-PD-L1 (Bio X Cell #BE0101) was delivered by I.P. at 0.2 mg in 0.2 mL per injection. For experiments involving anti-OX40 and/or anti-PD-L1, the control group was treated with the respective isotype antibodies (Bio X cell #BE0290, #BE0090). For experiments involving CpG, the control group was given saline. MHC-I on tumor cells were examined at 48 h post-CpG injection. Purified, endotoxin tested CpG oligonucleotides were ordered from Integrated DNA Technologies (IDT) with the following sequence: T*C*G*T*C*G*T*T*T*T*C*G*G*C*G*C*G*C*G*C*C*G.

Tumors were measured with digital calipers (Fisher Scientific). Tumor volumes were calculated as (length) ÷ 2 × (width)². Ethical endpoint was defined as tumor length of 20 mm in any direction. The maximal tumor size was not exceeded.

**Flow cytometry**. Each tumor was minced with a clean blade and then dissociated in 5 mL RPMI (Gibco) containing 1 mg/mL collagenase II (Gibco) for immunophenotyping or collagenase IV (Gibco) for MHC-I, 40 μg/mL DNase (Roche), 2% heat-inactivated FBS (Gibco), and 10 mM HEPES at 37 °C with constant agitation of 180 rpm for 25 min. Digested tissue was diluted with 30 mL of cold PBS and poured through a 70-micron nylon mesh strainer (Fisher) for tumor cell analyses or 40-micron nylon mesh strainer for immunophenotyping. Cells were pelleted at 220-300 x g and resuspended in 5 mL RBC lysis buffer (BioLegend) at room temperature. After 5 min, the cells were diluted with 25 mL of FACS buffer (HBSS with 1 mM EDTA and 2% heat-inactivated FBS), pelleted, and resuspended in <3 mL of PBS for cell counts. Cells were stained with the antibody panel for 30 min on ice, covered. Cells were washed with PBS and stained with fixable Near-IR live/dead stain (Molecular Probes) at 1:1,000 for 15 min at room temperature. For FOXP3 and Granzyme B staining, cells were fixed and permeated with a transcription factor staining buffer set (Invitrogen 00-5523-00) following staining of extracellular proteins. Cells were washed and resuspended in FACS buffer for data collection on a BD Dual Fortessa, using BD FACSDiva software (v 8.0.1), and analyzed with FlowJo (v 10.8.0). For MTB/TOM cells grown in culture, the same reagents and protocols were used after cells were lifted off the plate with a cell lifter (Corning), but the RBC lysis step was omitted. All experiments were compensated with single color controls and gating was determined by full panel minus one antibody or isotype antibody controls. All antibodies, unless otherwise indicated, were used at 1:100. A list of antibodies is in Supplementary Table 2.

**Histology and immunohistochemistry (IHC)**. Tissues were fixed in paraformaldehyde (Electron Microscopy Sciences, #15700), diluted to 4% with PBS, for 16–20 h and then moved to 70% ethanol. Samples were further processed by HistoWiz Inc, using their standard operating procedure and automated immunohistochemistry staining workflow with their in-house validated antibody list (published June 2018 [https://app.histowiz.com/ihc-antibodies], Supplementary Table 2).

**Immunohistochemistry quantification**. For each tumor and stain, three representative fields taken at 40X were imported for analysis using Fiji/ImageJ

(v 2.1.0/1.53) tool for color deconvolution for H DAB. The brown channel (color-2) was further selected for analysis. The threshold was standardized for each staining and set for all tumor samples. For T cell specific stains (CD3, CD4, CD8, FOXP3), H DAB positive particle counts were recorded. For F4/80 specific staining, the percentage of H DAB positive area (output % Area) was recorded.

**Western blot**. Cells were grown in 6-well dishes and grown to 90% confluence on the day of harvest. Cells or tumors were quickly washed once with 1 mL of cold PBS and lysed with Laemmli buffer (60 mM Tris-HCl, pH 6.8, 1 μM DTT, 2% w/v SDS) supplemented with protease inhibitor cocktail (Roche) and phosSTOP (Roche) or RIPA buffer (Thermo) supplemented with protease inhibitor cocktail and phosSTOP. Tumors were dissociated on ice in RIPA buffer supplemented with protease inhibitor cocktail and phosSTOP using a dounce homogenizer. DNA was sheared by passing lysates through a syringe or by water bath sonication at 4 °C. Protein extracts were quantified with DC Protein Assay (Bio-Rad) and prepared with NuPage sample loading buffer and reducing agent (Invitrogen). Proteins were resolved with the Bolt 4–12% Bis-Tris gel and buffer system (Invitrogen) and transferred onto nitrocellulose membranes using iBlot2 (Invitrogen) on program P0 for all proteins, except for B2M, which was transferred at 20 V for 6 min. Membranes were blocked with 5–10% non-fat milk (Rockland) in TBST and incubated with primary antibodies diluted in 5% non-fat milk TBST overnight on a 4 °C shaker. Membranes were washed with TBST and incubated with horseradish peroxidase (HRP)-conjugated secondary antibodies in TBST the next day. Signals were captured with ECL Prime (GE) or Visualizer (Millipore) on a Bio-Rad GelDoc system and Image Lab software (v 3). Images were exported to ImageJ (v 1.53) for quantification using 'Analyze > Gels'. The following antibodies were used: Anti-ß-actin (1:10,000, sc-47778 HRP, Santa Cruz Biotechnology), anti-c-MYC (1:1,000, clone Y69, ab32072, Abcam), anti-STAT1 (1:1,000, #9172, Cell Signaling), anti-B2M (1:10,000, ab75853, Abcam), Anti-Rabbit IgG (1:10,000, #7074, Cell Signaling). Uncropped and unprocessed scans of blots are in the Source Data file.

**RT-qPCR**. Cells were seeded on 6-well dishes and grown to 90% confluence on the day of harvest. Cells were quickly washed once in cold PBS, and total RNA was isolated using Trizol (Invitrogen) per manufacturer's instructions. 2 μg of RNA was used for cDNA synthesis using High-Capacity RNA-to-cDNA kit (Invitrogen). qPCR was completed with PowerUp SYBR Green Master Mix (Invitrogen) according to manufacturer protocol on a QuantStudio 6 Real Time PCR system (Applied Biosystems) and fold change was calculated using $\Delta\Delta C_t$ on the Quant-Studio Real Time PCR Software (v 1.7.1). Primers for cDNA are listed in Supplementary Table 3 and Supplementary Table 4.

### Patient data gene expression analysis and statistics

*TCGA*. The batch-corrected, RSEM-normalized gene level RNA-seq data from the 2018 TCGA Pan-Cancer Altas publications was used our analysis. Data was log2 transformed; and hallmark gene sets[32] were mapped to the dataset by gene symbol and scored using ssGSEA as implemented in the GSVA R package[68]. Clinical annotations from 1100 TCGA breast samples was obtained from the cBioPortal; and used to filter the expression dataset for breast samples. Expression data was log2 transformed and median-centered; and signature genes were mapped to the dataset by gene symbol. The MYC_BC (Supplementary Data 1) and JEM[16] MYC signatures were computed as the Pearson correlation to the directionality vector of the MYC signature genes (+1 for genes upregulated by MYC and −1 for genes downregulated by MYC); and immune cell signatures[33] were calculated as the mean of the signature genes. Pearson correlation between the MYC_BC signature and other signatures were assessed among the 158 TNBC samples based on ER and PR status by IHC and HER status by IHC or FISH (HER2-positive if either IHC or FISH is positive).

*ISPY-2*. We computed the MYC signature score from platform-corrected, normalized, log2-scaled, median-centered pre-treatment expression data (assayed on custom Agilent 44 K arrays). Signature genes were mapped to the dataset using gene symbol; and the MYC_BC signature was calculated as the Pearson correlation to the directionality vector of signature genes. A cut-point of 0 was used to dichotomize patients into High (>0) vs. Low (≤0) MYC_BC signature groups. Event-free survival was computed as time between treatment consent to loco-regional recurrence, distant recurrence, or death; and patients without event were censored at time of last follow-up. Cox proportional hazard modeling was used to assess the association between the MYC signature and event-free survival (EFS) in the 28 TNBC patients from the pembrolizumab arm with available follow-up data;[37] and Kaplan–Meier curves were constructed for visualization. All expression data analyses were performed using R (v 3.6.3).

*METABRIC*. METABRIC RNA microarray data was processed as described previously[35]. The $\log_2$ intensity of each of the MYC signature genes was median centered and the MYC signature was calculated as the Pearson correlation to the directionality vector of the MYC signature genes. We calculated the Spearman's correlation between the MYC signature and the $\log_2$ intensity of *B2M*, *NLRC5* and *CD274*. Additionally, we scored IL2/STAT5 signaling in METABRIC using ssGSEA (v2.0) and the hallmark IL2/STAT5 signaling gene set from MSigDB. Similarly, the

IL2/STAT5 signaling score was compared against the MYC signature using Spearman's correlation. Finally, we calculated a tumor infiltrating lymphocyte (TIL) signature as the mean of the 60 genes identified by Danaher et al[33]. and compared the TIL signature against the MYC signature using Spearman's correlation. All analyses were conducted considering triple negative breast tumors defined by hormone receptor expression.

*MYC signature in TONIC*. RNA sequencing of TNBC metastases at trial baseline (*n* = 53) and post-induction (*n* = 44) were generated as described previously[36]. RNA sequencing was aligned with STAR and abundance quantified by RSEM as implemented in the nextflow (v 20.12.0) pipeline nf-core/rnaseq (v 3.0)[69]. The TPM of each of the MYC genes was median centered and the MYC signature was calculated as the Pearson correlation to the directionality vector of the MYC signature genes, considering mRNA abundance at baseline. We compared the MYC signature between responders and non-responders post-induction using a logistic regression model correcting for induction strategy.

**Cell culture**. MTB/TOM cells were cultured in sterile conditions using DMEM with pyruvate (Gibco 11995065) supplemented with 10% FBS (Gibco) and 1% penicillin/streptomycin (UCSF Cell Culture Facilities) in 5% CO$_2$ at 37 °C as previously described[30]. Cells were maintained in the MYC-ON state with 1 μg/mL of doxycycline (Fisher #BP2653-5) in the media, and media was changed every 2 days. MCF10A-vector (puromycin) and MCF10A-MYC cells were previously published[54,70] and cultured in DMEM/F12 containing 5% horse serum, 20 ng/ml EGF, 0.5 mg/ml hydrocortisone, 100 ng/ml cholera toxin, 10 mg/ml insulin and 1X penicillin–streptomycin in 5% CO$_2$ at 37 °C[71]. MC38 cell line was a gift from the Spitzer lab at UCSF and was cultured in DMEM with pyruvate (Gibco 11995065) supplemented with 10% FBS (Gibco) and 1% penicillin/streptomycin (UCSF Cell Culture Facilities) in 5% CO$_2$ at 37 °C. MC38 cells were transduced with a retroviral vector containing human MYC and a hygromycin resistance gene or a vector only (with hygromycin resistance) and placed on selection for 2 weeks. All cells used in this study continuously tested negative for mycoplasma by PCR. For experiments with interferons, cells were seeded on day 1, treated with interferons on day 2, refreshed with new media and interferons on day 4, and harvested on day 5 (for a total of 72 h of treatment). IFNα was used at 1,000 U/mL (PBL Assay Science, #12115-1), IFNβ was used at 1,000 U/mL (PBL Assay Science, #12405-1), IFNγ was used at 100 ng/mL (Gibco, #PMC4031).

**Statistics for biological experiments**. Graphs and analysis performed in Prism 9 (v 9.1.0). Two-tailed, unpaired t-test was used for comparison between treatment groups in qPCR data, PD-L1 flow cytometry, tumor volumes, and immunohistochemistry quantification. For non-normally distributed data, such as flow cytometry experiments in animals, Mann–Whitney test was used. For Kaplan–Meier survival analysis in animals, log rank test was used. Outliers were determined for flow cytometry data. All animals that met the tumor enrollment criteria were included in our analysis.

**Reporting summary**. Further information on research design is available in the Nature Research Reporting Summary linked to this article.

## Data availability

No new datasets were generated in this study. The MTB/TOM and MYC-driven lymphoma RNA-seq datasets were previously analyzed and published[30] and downloaded from the Gene Expression Omnibus (GEO) repository under GSE130922. The RNA-seq for the MYC-driven liver model was previously analyzed and published[55] and downloaded from the GEO repository under GSE76078. We received permission to present de-identified participant I-SPY 2 data and since restrictions apply to the availability of these data, please contact the I-SPY 2 TRIAL Scientific Program Manager (ispyadmin@ucsf.edu) for access to the data. The TONIC Trial and METABRIC gene expression data are available on the European Genome-phenome Archive (EGA) under accession number EGAS00001003535 and EGAS00000000083, respectively. For gene ontology, we accessed the Molecular Signatures Database website (v 6.4) [http://www.gsea-msigdb.org/gsea/msigdb/annotate.jsp]. Patient outcomes for IMvigor210 Trial and metastatic ccRCC are available on the Tumor Immune Dysfunction and Exclusion (TIDE) database [http://tide.dfci.harvard.edu/query/]. The remaining data are available within the Article, Supplementary Information or Source Data file. Source data are provided with this paper.

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

## Acknowledgements

This work was supported by the US National Institutes of Health 1R01CA223817 (A.G.), F32CA243548 (J.V.L.), T32CA108462 (to J.V.L.); the CDMRP Breast Cancer Research Program W81XWH-16-1-0603 (A.G.) and W81XWH-21-1-0774 (A.G., J.K.); the Breast Cancer Research Foundation (H.S.R.); METAvivor Research Award (A.G.); The Mark Foundation Endeavor Awards (A.G.); The Gazarian Family Endowment (A.G.) and the National Science Foundation #1650113 (to R.N.). We acknowledge the PFCC (RRID:SCR_018206) for assistance generating flow cytometry data. Research reported here was supported in part by the DRC Center Grant NIH P30 DK063720. We thank all of the patients who volunteered to participate in the I-SPY 2 TRIAL.

## Author contributions

J.V.L., F.H., M.M., and A.G. designed the concept of the manuscript. Experiments were completed by J.V.L., R.N., J.W., F.H., J.M.A., P.M.M., M.S., D.V.d.M., G.H., G.A.H., and Y.Z. J.V.L. and F.H. completed the statistical analysis for biological experiments. C.Y. and K.E.H. completed and interpreted bioinformatic analysis of patient data. A.G., M.M., J.K., and C.C. supervised the studies. L.J.V., H.S.R., and L.J.E. were involved in clinical trial design and interpreted I-SPY 2 results. S.S. provided advocate perspectives to inform the work. J.V.L. wrote the original manuscript, and A.G., M.M., D.V.d.M., C.Y., H.S.R., R.N., S.S., C.B., and J.W. edited the text in this manuscript. All authors read the manuscript and agreed to submission of the manuscript for publication. M.K. generated the TONIC trial dataset and advised on these data.

## Competing interests

The authors declare no competing interests.
