## [Peer Review File · Nature Communications]

Reviewers' Comments:

Reviewer #1:

Remarks to the Author:

The study by Lee et al describes an important system of immunotherapy for MYC expressing tumors.

However, the structure format and style of the manuscript needs editing prior to being ready for scientific review.

For example:

1-The paper lacks headings for the abstract and the results sections making those sections blur into one another.

2-Instead of providing information about the background, motivation, and the intention of the study, the authors preemptively summarize their entire results in the introduction.

3-The figures are not presented in a logical and consecutive order. For instance the results section begins with figure 1g-i and then it jumps to figure 1b,a. In addition Figures 1cs-f are discussed only in the introduction and not in the results section.

4-There is no subheading to separate the results from figure 1 and figure 2.

Once these stylistic deficiencies and logical information flow are corrected the manuscript might be appropriate for scientific review, but not until.

Reviewer #2:

Remarks to the Author:

The manuscript by Lee et al describes that poor prognosis MYC-elevated triple negative breast cancer are resistant to checkpoint blockade in an inducible MYC-expressing mouse model of TNBC. The authors validate this finding in multiple clinical trials and show that in TNBC patients that have elevation of MYC, this is due to decreased TILs and MHC-1 expression. The authors then identify that combining a TLR9 agonist and an agonistic antibody against OX40 with anti-PD-L1 therapy results in enhanced tumour regression.

Major comments

1. How were the TILs assessed in the TCGA dataset? This is not clear from the methods. This would be useful to do using the Salgado score (as a gold standard in breast cancer) as well as assessment at GEX level and can be achieved through the available images in the TCGA portal. Are there any specific spatial distribution of TIL's in MYC-enriched TNBC's vs non?

2. Were there any differences in TNBC subtypes with MYC-elevation? i.e. immune response has been reported to be associated with the immune-modulatory subgroup of TNBC. What was the distribution of MYC-enriched TNBC's in regard to the subtypes versus non-MYC-elevated?

3. Are the authors able to corroborate the findings of decreased TILs and associations with MHC-1 etc in MYC-elevated TNBC additional TNBC cohorts such as METABRIC/SCAN-B as well?

4. The authors show that triplet combination of CpG/aOX40 and anti-PDL-1 therapy results in tumour regression in 75% of the animals treated. Do the authors have a reason for the no-response in the remaining 25% of animals? Did they look at T cell infiltrate, presence of T-regs etc? This may be informative going forwards as biomarkers of response using biopsies during treatment.

5. Are the authors able to validate this triplet combination in an additional MYC-elevated model? How efficacious is this in the non MYC-elevated models? This will help ascertain if indeed as the authors postulate this is indeed a MYC driven phenomenon.

Reviewer #3:

Remarks to the Author:

The study presented in the manuscript "Combinatorial immunotherapies overcome MYC-driven immune evasion" by Lee et al. is relevant to the important issue of tumor immune evasion and immunotherapy resistance in TNBC. Patients with recurrent metastatic TNBC have limited therapeutic options.

The major claims of the paper are that MYC signaling facilitates immune evasion in TNBC by inhibiting MHC-I expression on tumor cells and that combination therapy with TLR9 agonist CpG, anti-OX40, and anti-PD1 can overcome this evasion. The association between MYC, MHC I, and tumor immune evasion has been established previously (See PMID: 3402430 and PMID: 3096575 for select publications on a topic from 1986 and 1988). This manuscript highlights the relevance of this axis in the context of TNBC making it of interest to a specialized audience. The novel drug combination is interesting, however, the rationale for selecting these specific agents is not well explained and the mechanism of action is not explored. Below are few specific recommendations for the authors:

Figure 1A and 1C:

- The claim is that MYC-ON tumors are resistant to anti-PD-L1, while MYC-OFF tumors are sensitive. Tumor growth plot is provided for MYC-ON mice, and survival is shown for MYC-OFF. Both tumor growth and survival plots should be provided for MYC-ON and MYC-OFF mice on the same figure to allow direct comparison.
- What is the clinical relevance of starting anti-PD-L1 therapy on tumors that are as large as 10 mm in diameter? If patient presents with an enormously large primary breast tumor (considering the differences in body size of mice and humans), it is likely to be surgically removed before anti-PD-L1 therapy. See also next point on size.
- Are MYC-ON tumors truly resistant to anti-PD-L1 or are they simply too large in this particular experiment to be cleared by the endogenous immune cells? In fact, same model seems to respond to anti-PD-L1 in Fig. 4H (albeit no statistical analysis is provided). Visually, the start of the growth curves suggests that tumors were small at the beginning of treatment (starting size is not indicated in the legend). Based on Extended data 1d, turning MYC OFF caused dramatic shrinking of the tumors, so they were a lot smaller during the anti-PD-L1 therapy as compared to MYC-ON tumors. This argues that size of tumor may affect anti-PD-L1 response. The direct involvement of MYC signaling is not sufficiently illustrated.
- Figure 1B: The analysis is done in MYC-ON tumors only. PD-L1 expression on tumor and immune cells should be shown in MYC-OFF tumors as well.

Fig. extended data 1. What is the mechanism of tumors shrinking after turning MYC OFF? IHC for Ki67, and apoptotic markers will be helpful. It would be important to test if shrinking is immune dependent. T cell depletion experiments can be performed to test this.

Fig. 2h: Addition of flow cytometry analysis of tumor immune infiltrate and T cell phenotypes similar to Fig. 4d-g can make the point of MYC-associated immune evasion more convincing. The mechanism of reduced CD8 T cell levels in MYC-ON tumors is not clear. Is it dependent on tumor MHC-I?

Figure 4A: CpG induces MHC-I expression on tumor cells. Is this dependent on MYC inhibition or activation of the interferon signaling? Functional experiments using knockout/inhibitors are needed. Include levels of MYC and interferons in tumors of mice treated with and without CpG.

Fig. 4b: The rationale for adding anti-OX40 therapy is not fully clear. Results section states that CpG is known to induce OX40 on CD4 T cells. This should be confirmed in the studies model using flow cytometry. Also, a CD4 T cells depletion group can be added to the experiment shown in Fig. 4i to determine if these cells play role in response to studied triple combination.

Figure 4H: Provide average tumor growth curves and statistical comparison between treatment groups.

Statistical comparison is missing on Fig. 1A, 4C, 4H. Not clear what groups are compared in Fig. 4i.

Discussion: it would be good to highlight the state of clinical development of CpG and anti-OX40 so that the readers can understand the potential for clinical translation of studies drug combination.

RESPONSE TO REVIEWERS:

REVIEWER #1 (Remarks to the Author):

The study by Lee et al describes an important system of immunotherapy for MYC expressing tumors. However, the structure format and style of the manuscript needs editing prior to being ready for scientific review.

For example:

- 1-The paper lacks headings for the abstract and the results sections making those sections blur into one another.
- 2-Instead of providing information about the background, motivation, and the intention of the study, the authors preemptively summarize their entire results in the introduction.
- 3-The figures are not presented in a logical and consecutive order. For instance the results section begins with figure 1g-i and then it jumps to figure 1b,a. In addition Figures 1cs-f are discussed only in the introduction and not in the results section.
- 4-There is no subheading to separate the results from figure 1 and figure 2.

Once these stylistic deficiencies and logical information flow are corrected the manuscript might be appropriate for scientific review, but not until.

RESPONSE: We appreciate that the reviewer noted that our manuscript "describes an important system of immunotherapy for MYC expressing tumors" but were surprised that they were unable to provide a detailed review of our work. In the revised manuscript we have changed the format including an abstract, introduction and discussion sections, etc. We have also made sure that the figures are labeled consecutively and contain the appropriate headings, etc.

REVIEWER #2 (Remarks to the Author):

The manuscript by Lee et al describes that poor prognosis MYC-elevated triple negative breast cancer are resistant to checkpoint blockade in an inducible MYC-expressing mouse model of TNBC. The authors validate this finding in multiple clinical trials and show that in TNBC patients that have elevation of MYC, this is due to decreased TILs and MHC-1 expression. The authors then identify that combining a TLR9 agonist and an agonistic antibody against OX40 with anti-PD-L1 therapy results in enhanced tumour regression.

REVIEWER #2: 1. How were the TILs assessed in the TCGA dataset? This is not clear from the methods. This would be useful to do using the Salgado score (as a gold standard in breast cancer) as well as assessment at GEX level and can be achieved through the available images in the TCGA portal. Are there any specific spatial distribution of TIL's in MYC-enriched TNBC's vs non?

RESPONSE: We added text in the body of the manuscript to describe the paper that derived the TIL signature and how this was applied to the TCGA dataset to address the reviewer's question. Regarding the second question on the spatial distribution of TILs in MYC-enriched TNBCs, this is interesting, but we feel this is beyond the scope of this manuscript which is focused on how to improve anti-tumor immune responses in MYC-overexpressing tumors. Detailed analysis of immune spatial organization +/- MYC are being planned with other research groups at UCSF and could be part of a future paper. However, we include images of CD3 staining in MTB/TOM MYC-ON and MYC-OFF tumors for the reviewer to evaluate (see below). These images indicate that some CD3 staining is on the outside (periphery) of the tumor in the MYC-ON (ON DOX) condition but more CD3 staining is present in the center of the tumor in MYC-OFF (OFF DOX) condition. Arrows point to examples of CD3+ cells (see figure included immediately below). Further spatial localization will require

additional staining to distinguish tumor cells from supporting stroma. However, this additional information, although interesting will not further the conclusions of the current manuscript.

REVIEWER #2: 2. Were there any differences in TNBC subtypes with MYC-elevation? i.e. immune response has been reported to be associated with the immune-modulatory subgroup of TNBC. What was the distribution of MYC-enriched TNBC's in regard to the subtypes versus non-MYC-elevated?

RESPONSE: We looked at the paper published by Lehmann et al and find that MYC is associated with the BL1 TNBC subtype, which is characterized by cell cycling and proliferation. MYC is not associated with the immune-modulatory subgroup of TNBC. We have added a sentence describing this in the **discussion section** of the manuscript.

REVIEWER #2: 3. Are the authors able to corroborate the findings of decreased TILs and associations with MHC-1 etc in MYC-elevated TNBC additional TNBC cohorts such as METABRIC/SCAN-B as well?

RESPONSE: In addition to the TCGA cohort, in the revised manuscript we also examined the MYC signature in the large METABRIC TNBC cohort and in the TONIC trial, which show similar correlations as observed in the TCGA TNBC cohort. These corroborating findings further support the association between the newly defined MYC breast cancer signature (MYC_BC) and diminished immune infiltration and downregulation of the MHC-I pathway. These new data are included in the revised manuscript (**Figure 1k, l; Figure 2b, d; Figure 3 b, c; Extended Data Figure 4, 5**).

REVIEWER #2: 4. The authors show that triplet combination of CpG/aOX40 and anti-PDL-1 therapy results in tumour regression in 75% of the animals treated. Do the authors have a reason for the no-response in the remaining 25% of animals? Did they look at T cell infiltrate, presence of T-regs etc? This may be informative going forwards as biomarkers of response using biopsies during treatment.

RESPONSE: Repeat longitudinal biopsies of the same tumor would be the ideal approach to determine biomarkers of response. However, collecting biopsies is not technically possible in the mouse models as it is

done in some clinical trials. Furthermore, samples gathered of tumors at endpoint (when it is possible to identify the non-responding tumors which have grown out) would be difficult to interpret due to massive tumor necrosis and lack of an appropriate comparator group. This is a limitation to the study. We plan future studies to serially passage the 25% of tumors non-responsive to triple-therapy to determine if the causes are due to tumor-intrinsic effects (i.e. activation of new oncogene signaling pathways, silencing of MHC-I, etc) or alterations of the host immune system. While important we feel that this would be an entirely new direction and beyond the scope of the present study.

REVIEWER #2: 5. Are the authors able to validate this triplet combination in an additional MYC-elevated model? How efficacious is this in the non MYC-elevated models? This will help ascertain if indeed as the authors postulate this is indeed a MYC driven phenomenon.

RESPONSE: We have approached this by testing the MC38 tumor model which is sensitive to anti-PD-L1 therapy (Juneja, et al 2017). We overexpressed MYC in this model and find that it is no longer sensitive to anti-PD-L1 monotherapy (**Figure 1 c, d**). Additionally, we have demonstrated that the MC38-MYC tumors are responsive to the triple therapy. These new data are incorporated in the manuscript (**Figure 4k**). We also worked with a collaborator in Finland, who tested another MYC-driven breast cancer model (WAP-MYC) in their facility. The improved response with triplet therapy were reproducible in the hands of another independent laboratory and animal facility, which further supports the effectiveness of the combination immunotherapy for MYC-elevated tumors (**Figure 4l**). We thank the reviewer because these new data help to further strengthen our conclusions.

However, it was not our intention to convey that the triple-therapy would only work in MYC-elevated tumors. Our work serves to demonstrate that MYC imposes challenges to successful immunotherapy in patients with MYC-elevated tumors, thus additional therapeutics that stimulate interferon response in combination with anti-PD-L1 will likely improve survival benefit.

REVIEWER #3 (Remarks to the Author):

The study presented in the manuscript “Combinatorial immunotherapies overcome MYC-driven immune evasion” by Lee et al. is relevant to the important issue of tumor immune evasion and immunotherapy resistance in TNBC. Patients with recurrent metastatic TNBC have limited therapeutic options.

The major claims of the paper are that MYC signaling facilitates immune evasion in TNBC by inhibiting MHC-I expression on tumor cells and that combination therapy with TLR9 agonist CpG, anti-OX40, and anti-PD1 can overcome this evasion. The association between MYC, MHC I, and tumor immune evasion has been established previously (See PMID: 3402430 and PMID: 3096575 for select publications on a topic from 1986 and 1988). This manuscript highlights the relevance of this axis in the context of TNBC making it of interest to a specialized audience. The novel drug combination is interesting, however, the rationale for selecting these specific agents is not well explained and the mechanism of action is not explored. Below are few specific recommendations for the authors:

Figure 1A and 1C:

· The claim is that MYC-ON tumors are resistant to anti-PD-L1, while MYC-OFF tumors are sensitive. Tumor growth plot is provided for MYC-ON mice, and survival is shown for MYC-OFF. Both tumor growth and survival plots should be provided for MYC-ON and MYC-OFF mice on the same figure to allow direct comparison.

RESPONSE: We have added the survival plot for the MYC-ON mice to complete the panel (**Figure 1a, b, e, f**).

REVIEWER #3: · What is the clinical relevance of starting anti-PD-L1 therapy on tumors that are as large as

10 mm in diameter? If patient presents with an enormously large primary breast tumor (considering the differences in body size of mice and humans), it is likely to be surgically removed before anti-PD-L1 therapy. See also next point on size.

· Are MYC-ON tumors truly resistant to anti-PD-L1 or are they simply too large in this particular experiment to be cleared by the endogenous immune cells? In fact, same model seems to respond to anti-PD-L1 in Fig. 4H (albeit no statistical analysis is provided). Visually, the start of the growth curves suggests that tumors were small at the beginning of treatment (starting size is not indicated in the legend). Based on Extended data 1d, turning MYC OFF caused dramatic shrinking of the tumors, so they were a lot smaller during the anti-PD-L1 therapy as compared to MYC-ON tumors. This argues that size of tumor may affect anti-PD-L1 response. The direct involvement of MYC signaling is not sufficiently illustrated.

RESPONSE: We started the animals on treatment at 1 cm because that is the size the tumors are irrefutably aggressive and growing. We understand the reviewer's concern that this tumor might be too big to respond to therapy. We previously completed this experiment with tumors starting at 0.5 cm and 1 cm and saw similar results, suggesting that tumor size does not influence lack of response to anti-PD-L1 monotherapy (see below).

In Figure 4 of the manuscript, we initiated treatment at 0.5 cm. The starting size is indicated in the text and the complementary figures (Extended data figure 1b, c; Extended data figure 10).

Further, we approached this by testing the MC38 colon carcinoma cells in C57BL/6 mice, which was previously demonstrated to be sensitive to anti-PD-L1 (Juneja, 2017). We demonstrate in the MC38 model that when we start anti-PD-L1 at 0.5 cm, tumor growth is slowed in the MC38-control (vector) line, but anti-PD-L1 is no longer effective in the MC38-MYC overexpressing line, demonstrating that the response is due to MYC rather than size of the starting tumor. The new data are added to the paper (Figure 1 c, d; Extended Data Figure 1c).

REVIEWER #3: · Figure 1B: The analysis is done in MYC-ON tumors only. PD-L1 expression on tumor and immune cells should be shown in MYC-OFF tumors as well.

RESPONSE: We analyzed PD-L1 expression by flow cytometry on MYC-OFF tumors and find that PD-L1 expression does not change on the tumor cells but there is a slight increase in CD11b+, Ly6C- myeloid cells. The new data are added to the paper (Extended Data Fig. 1e).

REVIEWER #3: Fig. extended data 1. What is the mechanism of tumors shrinking after turning MYC OFF? IHC for Ki67, and apoptotic markers will be helpful. It would be important to test if shrinking is immune dependent. T cell depletion experiments can be performed to test this.

RESPONSE: We have included H&E images and staining for Ki67 in MYC-ON and MYC-OFF tumors (Extended Data Fig. 1 and pasted below). We also tested whether immune cells are required for tumor shrinking during

the MYC-OFF state by growing the tumors in NSG mice lacking mature B, T and NK cells. We conclude that tumor shrinking during the MYC-OFF state is not dependent on having a functional adaptive immune system or NK cells (see below).

REVIEWER 3: Fig. 2h: Addition of flow cytometry analysis of tumor immune infiltrate and T cell phenotypes similar to Fig. 4d-g can make the point of MYC-associated immune evasion more convincing. The mechanism of reduced CD8 T cell levels in MYC-ON tumors is not clear. Is it dependent on tumor MHC-I?

RESPONSE: We have now performed flow cytometry analysis of the tumor immune infiltrate and T-cell phenotypes in **figure 2i**. We see an increase in T-cell infiltration and a trend towards an increase in CD8+ T cells in the MYC-OFF state. There is also an increase in Granzyme B (GrB) expression in the CD8+ T cells in the MYC-OFF tumors. Regarding the question of MHC-I expression and T-cell infiltration, prior studies using immunocompetent models have not found an association between MHC-I expression and CD8+ T cell infiltration. For example, others have shown that *B2M* knock-out in MC38 cells does not lead to a change in tumor infiltrating CD8+ T cells (Torrejon et al, Cancer Discovery 2020). The new flow cytometry characterization of the MYC ON vs OFF tumors are further discussed in the revised manuscript.

REVIEWER #3: Figure 4A: CpG induces MHC-I expression on tumor cells. Is this dependent on MYC inhibition or activation of the interferon signaling? Functional experiments using knockout/inhibitors are needed. Include levels of MYC and interferons in tumors of mice treated with and without CpG.

RESPONSE: To examine the connection between interferon signaling and MYC abundance we performed the following *in vivo* experiments. Following intratumoral CpG injection, we collected tumors and checked for MYC expression by western blot. CpG is well-known to increase Type I interferon production *in vivo*, which occurs in our tumor samples as evidenced by an increase in STAT1 and p-STAT1 (Y701) signaling (**Extended Data Figure 10**). MYC levels are not diminished upon CpG treatment *in vivo*. Mechanistically, CpG induces MHC-I in tumors (Figure 4a), but CpG induction of MHC-I is independent of MYC expression.

REVIEWER 3: Fig. 4b: The rationale for adding anti-OX40 therapy is not fully clear. Results section states that CpG is known to induce OX40 on CD4 T cells. This should be confirmed in the studies model using flow cytometry. Also, a CD4 T cells depletion group can be added to the experiment shown in Fig. 4i to determine if these cells play role in response to studied triple combination.

RESPONSE: The rationale to add anti-OX40 came from the Dr. Levy group's paper (Sagiv-Barfi et al, *Science Trans. Med.*, 2018). They demonstrated that CpG induced OX40 on CD4+ T-cells. We also observed an increase in OX40 expression on CD4+ T-cells after CpG treatment (**See figure below**) in MYC-driven TNBC tumor models. In addition, we see that CpG+anti-PD-L1 was insufficient to elicit a survival benefit beyond CpG alone (**Extended Data Figure 11**), further bolstering the need for OX40 agonist. We expect CD4 and CD8 T-cells are both required for a survival benefit in response to CpG/aOX40 treatment.

REVIEWER 3: Figure 4H: Provide average tumor growth curves and statistical comparison between treatment groups.

Statistical comparison is missing on Fig. 1A, 4C, 4H. Not clear what groups are compared in Fig. 4i.

RESPONSE: We have now added the statistical information to the figure legend for these figures. We chose to show the individual tumor growth curves in Figure 4 to demonstrate that some animals respond to therapy but others do not. We feel that showing the average growth curves does not accurately convey the information since some mice experienced full tumor shrinkage but not others. Nevertheless, we have attached the average growth curves for the reviewers to evaluate (see figure below). These demonstrate the improved response observed with triple-therapy. We've also included the average tumor volumes at the last day of treatment (day 19) in the manuscript for statistical comparison (new Figure 4i).

REVIEWER 3: Discussion: it would be good to highlight the state of clinical development of CpG and anti-OX40 so that the readers can understand the potential for clinical translation of studies drug combination.

RESPONSE: We have now added the information on the ongoing clinical trials involving CpG and anti-OX40 to the discussion.

Reviewers' Comments:

Reviewer #2:

Remarks to the Author:

The authors have adequately addressed the concerns raised, however further clarification in the text would be helpful for some of the points.

1 It would be useful here to add a sentence in the discussion to point to the fact future work will focus on the spatial distribution of TILs in these tumors.

4. This limitation of the study should be added to the discussion.

Reviewer #3:

Remarks to the Author:

The authors have addressed my comments

RESPONSE TO REVIEWERS' COMMENTS

Reviewer #2 (Remarks to the Author):

The authors have adequately addressed the concerns raised, however further clarification in the text would be helpful for some of the points.

1 It would be useful here to add a sentence in the discussion to point to the fact future work will focus on the spatial distribution of TILs in these tumors.

Response: Added: "Future work to explore the spatial distribution of TILs in MYC-high TNBCs and following combinatorial immunotherapies should provide additional insight into mechanisms of immune evasion and how they can be overcome."

4. This limitation of the study should be added to the discussion.

Response: Added: "In future studies, it will be important to also understand the signaling pathways and biomarkers in the small subset of MYC-high tumors that did not respond to triple-combination immune therapies."

Reviewer #3 (Remarks to the Author):

The authors have addressed my comments